# Antihuman Endogenous Retrovirus Immune Response and Adaptive Dysfunction in Autism

**DOI:** 10.3390/biomedicines10061365

**Published:** 2022-06-09

**Authors:** Alessandra Carta, Maria Antonietta Manca, Chiara Scoppola, Elena Rita Simula, Marta Noli, Stefano Ruberto, Marta Conti, Ignazio Roberto Zarbo, Roberto Antonucci, Leonardo A. Sechi, Stefano Sotgiu

**Affiliations:** 1Unit of Child Neuropsychiatry, Department of Medical, Surgical and Experimental Sciences, University of Sassari, 07100 Sassari, Italy; alessandra.carta@aouss.it (A.C.); chiara.scoppola@gmail.com (C.S.); marta.conti7@gmail.com (M.C.); 2Unit of Microbiology and Virology, Department of Biomedical Sciences, University of Sassari, 07100 Sassari, Italy; m.anto.manca@gmail.com (M.A.M.); simulaelena@gmail.com (E.R.S.); martanoli@outlook.it (M.N.); ruberto.ste@gmail.com (S.R.); sechila@uniss.it (L.A.S.); 3Unit of Neurology, Department of Medical, Surgical and Experimental Sciences, University of Sassari, 07100 Sassari, Italy; robertozarbo@gmail.com; 4Pediatric Clinic, Department of Medical, Surgical and Experimental Sciences, University of Sassari, 07100 Sassari, Italy; rantonucci@uniss.it; 5Unità Complessa di Microbiologia e Virologia, AOU Sassari, 07100 Sassari, Italy

**Keywords:** human endogenous retroviruses, autism spectrum disorders, immune response, antibodies, adaptive behaviour

## Abstract

ASD is a neurodevelopmental disorder of unknown aetiology but with a known contribution of pathogenic immune-mediated mechanisms. HERVs are associated with several neuropsychiatric diseases, including ASD. We studied anti-HERV-W, -K and -H-env immune profiles in ASD children to analyse differences between their respective mothers and child/mother control pairs and possible correlations to ASD severity and loss of adaptive abilities. Of the 84 studied individuals, 42 children (23 ASD and 19 neurotypical) and their paired mothers underwent clinical and neuropsychological evaluations. ASD severity was analysed with standardised tests. Adaptive functioning was studied with ABAS-II and GAC index. Plasma anti-env responses of HERV-K, -H and -W were tested with indirect ELISA. ASD and neurotypical children did not differ in age, gender, comorbidities and anti-HERV responses. In children with ASD, anti-HERV levels were not correlated to ASD severity, while a significant inverse correlation was found between anti-HERV-W-248-262 levels and adaptive/social abilities. Upregulation of anti-HERV-W response correlates to dysfunctional social and adaptive competences in ASD but not in controls, suggesting anti-HERV response plays a role in the appearance of peculiar ASD symptoms.

## 1. Introduction

Autism spectrum disorder (ASD) is a persistent and disabling neurodevelopmental disorder characterised by deficits in social communication and interaction across multiple contexts, as well as restricted, repetitive patterns of behaviour, interests or activities [1]. The growing prevalence of ASD has been recently estimated to be at least 1.5% in developed countries, mostly among those patients without comorbid intellectual disability [2]. Although the lifelong worldwide prevalence of ASD has doubled during recent decades [3] and pooled environmental and genetic factors can clearly contribute to the ASD-risk increase, its aetiology remains not yet understood [4]. Moreover, identified biomarkers associated with clinical symptom severity and adaptive functioning in ASD may reflect the effects rather than the causes of autism [5].

Human endogenous retroviruses (HERVs) are complete or partial germline sequences remnant of exogenous retrovirus infections that integrated their genomes thousands of years ago [6]. After the discovery of the multiple sclerosis endogenous retrovirus (MSRV) [7], the neurogliotoxic properties of HERV-envelope (env) proteins [6] and their transactivation by infectious agents [8], a prolific era has raised associating HERVs with the development of central nervous system (CNS) diseases such as MS [9] or psychiatric conditions such as schizophrenia [10] and bipolar disorder [11]. Although of unclear significance, the expression of HERV-env families (H, K, W) from peripheral mononuclear cells has been found to also be skewed in patients with ASD [12]. Interestingly, increased HERV-H and reduced HERV-W-env mRNAs have been found in ASD samples compared to controls, particularly in patients with a severe ASD phenotype and a serum proinflammatory cytokine profile [13].

Immune dysregulation has known detrimental effects on foetal neurodevelopment. Maternal immunoglobulins and cytokines cross the placenta and enter the foetal compartment for protective purposes. However, owing to the immaturity of the developing CNS, harmful autoantibodies and cytokines can negatively impact brain homeostasis [14]. In animal models, maternal immune activation elicited by viruses, bacteria or lipopolysaccharides, or the administration of maternal CNS autoantibodies [15], can result in ASD-like manifestations in offspring or intellectual disability without ASD [16].

In our hands, abnormal serum antibody responses to HERV-W and HERV-K-env families were found in autoimmune diseases such as MS [17], rheumatoid arthritis [18] and type 1 diabetes [19]. Here, and for the first time, we studied anti-HERV-W, -K and -H-env immune responses in children with ASD. Our purposes were to analyse whether the anti-HERV profile of ASD children differs from their mothers and matched healthy child/mother pairs, whether it correlates to ASD severity, and the extent of the adaptive abilities of young subjects, regardless of ASD condition. To pursue this aim, we addressed three specific research questions:Is there any difference in the anti-HERV response family between the case and the control groups?Are anti-HERV immune responses associated with clinical ASD severity levels by measuring core symptoms, intellective ability and clinical global severity scales?Are anti-HERV antibody levels correlated with adaptive abilities and social competences measured with standardised questionnaires such as ABAS-II (GAC score)?

## 2. Materials and Methods

### 2.1. Enrolment Criteria

The study was approved by our local Ethics Committee (PG/2022/4977) and complied with the Declaration of Helsinki.

Patients with ASD, diagnosed according to DSM-5 criteria (APA, 2013), their mothers and control mother/child pairs accepted their enrolment by signing informed consent. All individuals were enrolled according to strict inclusion and exclusion criteria (Table 1).

The initial sample consisted of 128 subjects who met the inclusion criteria and agreed to take part in this study: 64 children with and without ASD diagnosis and their 64 paired mothers. Of these, 44 subsequently left the study: 17 cases due to the lack of collaboration in blood sampling and 14 because, after providing initial consent, they changed their opinion and dropped out of the study. Finally, 13 received a comorbid diagnosis immediately after study entry (3 children in the ASD group received a diagnosis of genetic syndrome, and 10 children of the control group had a suspicion of a neurodevelopmental condition after their clinical evaluations).

A final sample of 84 Caucasian individuals was collected: 42 children (23 with ASD diagnosis and 19 typically developing controls) and their 42 paired mothers (23 of the children with ASD and 19 of the 19 controls).

### 2.2. Clinical Evaluation and Standardised Instruments

Patients with ASD were subject to clinical and standardised evaluations, using both structured tests and parent questionnaires.

ASD core symptoms were evaluated with the *Autism Diagnostic Observation Schedule*, *2nd Edition* (ADOS-2—Lord, 2012) and the *Social Responsiveness Scale Questionnaire* (SRS—Constantino, 2005); quantification of clinical severity was conducted with the *Clinical Global Impression*—*Severity Scale* (CGI-S—Busner, 2007) and the *Children-Global Assessment Scale* (C-GAS—Shaffer, 1983), lead by a rater child and adolescent psychiatrist. Intellective evaluation was performed with verbal, WISC-IV (Wechsler, 2003), and nonverbal, Leiter-3 (Roid, 2013) cognitive tests, as well as the Raven progressive matrices (Raven, 1998). Adaptive functioning was studied with the *Adaptive Behavior Assessment System*, *Second Edition* (ABAS-II—Oakland, 2003) summarised by the global adaptive composite (GAC). Control children were analysed with CGI-S, C-GAS, ABAS-II (GAC index) and the SRS questionnaires to exclude the presence of clinical ASD symptoms.

The neuropsychological profile of mothers was studied with WAIS-IV (Wechsler, 2008) and unstructured anamnestic interviews led by clinicians, to rule out atypical neurodevelopmental conditions.

### 2.3. Blood Samples, Peptides and Enzyme-Linked Immunosorbent Assay (ELISA)

Plasma was isolated from peripheral venous blood samples and tested for the presence of antibodies against specific portions of the envelope protein of HERV-K, HERV-H and HERV-W. The immunogenic epitope prediction, related to the envelope sequence, was performed using the Immuno Epitope Database (IEDB). HERV-K-env (19–37), HERV-W-env (248–262) and HERV-H-env (229–241) epitopes (Table 2), synthesised at >95% purity, were dissolved in DMSO and stored at −80 °C. Samples were analysed according to our previously developed in-house indirect ELISA [17,18,19]. Results were expressed as the means of duplicates of 405 nm optical density (OD) values.

### 2.4. Statistical Analysis

Data were analysed using software R 3.4.1 and GraphPad Prism 8.2.0 software (San Diego, CA, USA). The Shapiro–Wilk normality test was used to analyse sample distribution. The Chi-square test was used for descriptive statistics and the ANOVA test for multivariable analysis. Pearson’s correlation was used to analyse the relationship between the global adaptive composite (GAC) index and the global ABAS-II score with anti-HERV antibody levels in ASD and neurotypical children. Linear regression analysis was used to test anti-HERV levels with the age of participants. Significance was set at *p* < 0.05.

Based on previous literature data, predicting that approximately 68% of children with ASD and their mothers and approximately 33% of typically developing children and their mothers will show significant greater expression of HERV-H and, to a lesser extent of HERV-W [12,13], we guaranteed an adequately enhanced study with a statistical power of 80% and an alpha error of 5%. For variables with missing data, the power was reduced. The study addressed three specific and preplanned research questions, which led our analyses towards well-defined hypotheses, based on previous reference literature. Therefore, no multiple testing correction was conducted, in line with specific guidelines [20].

## 3. Results

### 3.1. Demography

The sample was normally distributed according to the Gaussian bell of the Shapiro–Wilk test and did not show a significant departure from the normality for age, *W*(42), *p* = 0.076, nor for sex *W*(42), *p* = 1.000.

Patients with ASD (n = 23, 65% males) had a mean age of 6.5 years (range: 2–16). Ten (43.5%) had associated comorbidities (intellectual disability, attention-deficit with hyperactivity disorder, dyspraxia); eight (35%) had psychiatric comorbidities (depression, anxiety, obsessive-compulsive disorder), often in combination. Fourteen (61%) had a family history of neurodevelopmental disorders and seven (30%) of preterm birth with no significant complications.

Twenty-three healthy mothers of autistic subjects had a mean age of 43 years (range: 27–54). Spontaneous abortion was reported in 22% of them.

Nineteen neurotypical control children had a mean age of six years (range: 3–17). Two had specific learning disorders and four (20%) psychiatric comorbidities (depression, anxiety). Two had a family history of neurodevelopmental disorders and one of preterm birth without complications. Their 19 healthy mothers (mean age: 30.5 years, range: 26–52) had spontaneous abortions in 10.5%.

ASD and matched neurotypical children showed nonsignificant differences in terms of age, gender, psychiatric comorbidities and mothers’ mean age, except for other neurodevelopmental disorders, which were more frequently represented in the ASD sample, and prematurity, which was a more frequent complaint in the mothers of ASD children (Table 3 and Table 4).

### 3.2. Neuropsychiatric Features

Neuropsychiatric functioning was evaluated using ABAS-II, SRS, CGI-S and C-GAS scales. As expected, scores on global functioning were significantly lower for autistic subjects compared to controls in ABAS-II total score and subscores (*p* = 0.01). Global dysfunctioning (CGI-S and C-GAS) was significantly higher in ASD compared to controls (*p* < 0.001; Table 5)

Ten patients (43.5%) had Level 1 autism severity, seven (30.4%) had Level 2 and six (26.1%) Level 3. Intellectual disability was found in seven ASD patients (30.4%; two patients with Level 1 severity, two with Level 2 and three with Level 3). ABAS-II and SRS scores, along with CGI-S and C-GAS evaluations, were significantly worse in ASD compared to neurotypical controls (*p* < 0.001 for all; not shown).

### 3.3. Anti-HERV Profile

For all groups, the anti-HERV-H, -W and -K-env antibodies had a homogeneous profile (e.g., anti-H and anti-W: 0.67 in ASD and 0.85 in HC, respectively, Figure 1). Linear regression showed an inverse slope, although it did not reach statistical significance between each antibody level and the age of donors in all children and mothers (Figure 2 and Table 6).

### 3.4. Anti-HERV Responses to Address Our Three Specific Research Questions

#### 3.4.1. Is There Any Difference in Anti-HERV Family between the Case and Control Groups?

As for the research question, the ANOVA test indicates no significant differences between ASD, neurotypical controls and their respective mothers for all anti-HERV family immune responses (Figure 3 and Table 7).

#### 3.4.2. Are Anti-HERV Families Associated with Clinical ASD Severity Levels?

No correlation was found between anti-HERV levels and clinical ASD severity (Level 1–3) or the coexistence of intellectual disability (not shown).

#### 3.4.3. Are Anti-HERV Antibody Levels Correlated with Adaptive Abilities and Social Competences as Measured with ABAS-II (GAC Index)?

Anti-HERV antibody levels inversely correlated with adaptive abilities and social competences as measured with ABAS-II (GAC index) in children with ASD (Figure 4 and Table 8), but not in their neurotypical controls. In particular, the highest scores of ABAS-II-GAC were significantly correlated with the lowest level of anti-HERV-W-248-262 (Figure 4), but not anti-HERV-K responses. Other subscores of ABAS-II and SRS showed no correlation with the antibody levels.

## 4. Discussion

Several pieces of the etiopathogenic puzzle of ASD are still missing. A consensus has developed around ASD’s immune-mediated pathophysiology and the lifelong exposure of the ASD brain to innate immunity activation [21,22]. Other studies have tried to understand some ASD subtypes as a prototypical, maternal, antibody-mediated disease [23]. Nevertheless, the joint involvement of genetic and environmental factors and the influence of maternal infections during gestation is substantially undeniable in ASD pathogenesis [24]. However, the idea of ASD as a classical autoimmune disease is challenged by the multifaceted nature of many immune effectors, which can be either beneficial or detrimental depending on resilience factors to hostile events, tissue homeostasis, neurodevelopmental stage, inherited and de novo gene mutations, and many others [24,25] A derailed maternal–foetal neuroimmune crosstalk can initiate and maintain a chronic neuroglial activation, eventually causing complex alteration of neurogenesis, migration, synapse formation and, later in life, pruning [25]. Here, and to our knowledge for the first time, we found new evidence that is in line with modern views of ASD pathophysiology.

Firstly, despite a significant skewing of peripheral HERV activation in ASD has been documented [12,13], the immune response toward expressed HERV-env proteins does not differ between ASD subjects, their mothers and paired controls. We suggest that the adaptive immunity is not altered in ASD and that peripheral anti-HERV responses reflect mechanisms that are not linked to ASD aetiology.

Secondly, quantitative variation of anti-HERV loads and adaptive functions as measured with ABAS-II scores are significantly and negatively correlated. This relationship seems to be ASD-specific, owing to its absence in neurotypical children, and not influenced by ASD severity (Level 1–3) nor by a concomitant intellectual disability.

The high strength of this result could suggest that the variation in antibody load is clinically effective only in individuals that are already, and perhaps for reasons other than HERVs, predisposed to ASD development. In fact, in resilient, neurotypical controls, such an identical immune response is irrelevant to functional abilities [24]. Moreover, our sample was normally distributed and well powered. Finally, the methodological approach of our study included well-defined and stringent inclusion and exclusion criteria, to avoid interactions of results with nonspecific factors of ASD, which could have conditioned the immunological response to HERVs.

However, the limitations of this study are worth noting. First of all, a concomitant analysis of HERV gene expression was not performed. However, it is essential to remember how the expression of endogenous retroviruses is extremely vulnerable to a large number of variables: environmental, physiological and pathological changes (i.e., stressors).

The mechanism underlying the link between anti-HERV response and ASD social dysadaptation could be very entangled. In fact, although HERV proteins are physiologically expressed in normal human brain [26], little is known about their function. It is noteworthy to underline the deceitful nature of HERV antigens (Ags). Theoretically, the host’s immune system should recognise HERV antigens (Ags) as “self-Ags” since they are part of the human genome and, thus, immune tolerated. Notwithstanding, HERV Ags are able to trigger both innate and adaptative responses, probably because of their similarity with exogenous viral proteins.

In proinflammatory circumstances, the chronically activated microglia of ASD brain [21], the subsequent overexpression of HERV-W and -K family envelope proteins [8,27] and the activation of other immune cells and/or detrimental cytokines would perpetuate the endless loop of tissue damage [11]. Furthermore, it has been well described how HERVs can be regulated by epigenetic signals, which, in turn, can be activated by stressful or traumatic experiences such as social maladaptation in children and adolescents with ASD [28].

Here, we showed that the lower the anti-HERV-W antibody level, the better the adaptive functioning in subjects with ASD. From a clinical point of view, this is an interesting point, meaning that children and adolescents with ASD who showed the best adaptive global functioning are those who had the lowest HERV-W levels. This discovery leads to the conclusion that anti-HERV-W antibodies could be used as a stress-response biomarker in ASD, allowing for a significantly inverse correlation with specific clinical patterns such as global adaptive functioning.

## 5. Conclusions

Our scenario seems to be robust enough to support the hypothesis of a possible specific, though limited, role of some anti-HERV responses in the appearance of peculiar symptoms of ASD. Our outcome suggests a replication study to coevaluate it with the HERV gene expression and, since HERV-W levels could be associated with a stress response, with the inclusion of a high-stress, non-ASD control group, through a follow-up study design.

## Figures and Tables

**Figure 1 biomedicines-10-01365-f001:**
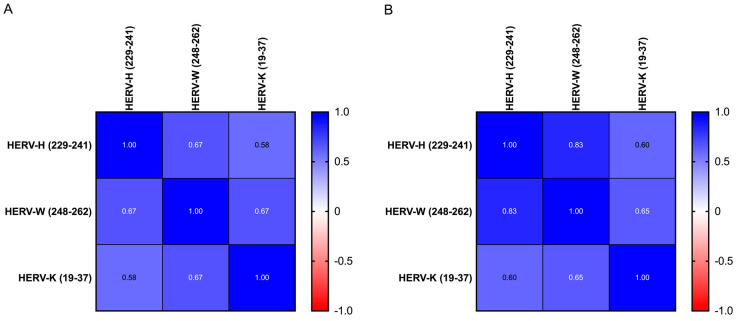
Heatmap displaying the r values obtained from Spearman’s correlation analysis performed among HERV-H-, HERV-W- and HERV-K-derived epitopes in ASD children (**A**) and in HC children (**B**).

**Figure 2 biomedicines-10-01365-f002:**
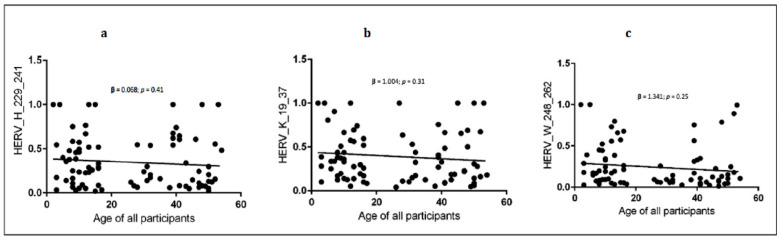
Linear regression between HERVs antibody levels and age of all participants (children and mothers). (**a**) shows the linear regression between anti-HERV-H and age; (**b**) shows the linear regression between anti-HERV-K and age; (**c**) shows the linear regression between anti-HERV-W and age.95% CI, confidence interval fixed at 95%. Statistical significant levels fixed at * *p* < 0.05.

**Figure 3 biomedicines-10-01365-f003:**
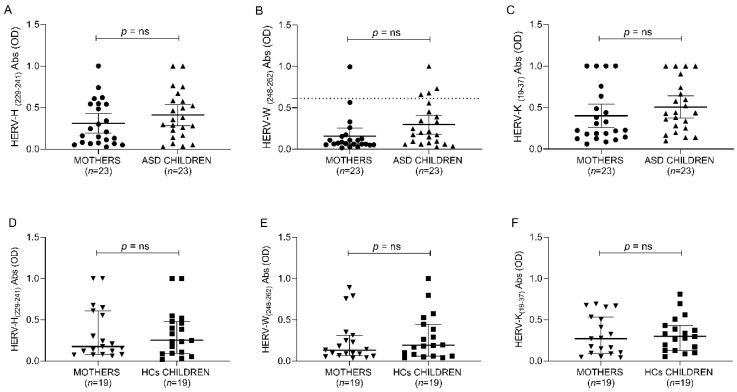
ELISA-based analysis of antibody presence against HERV-H-, HERV-W- and HERV-K-derived epitopes. Plasma samples from ASD and HCs populations were tested against HERV-H (229–241) (**A**,**D**), HERV-W (248–262) (**B**,**E**) and HERV-K (19–37) (**C**,**F**). Scatter plots represent median with 95% of CI. The Mann–Whitney *p*-value is indicated in the upper part of each graph.

**Figure 4 biomedicines-10-01365-f004:**
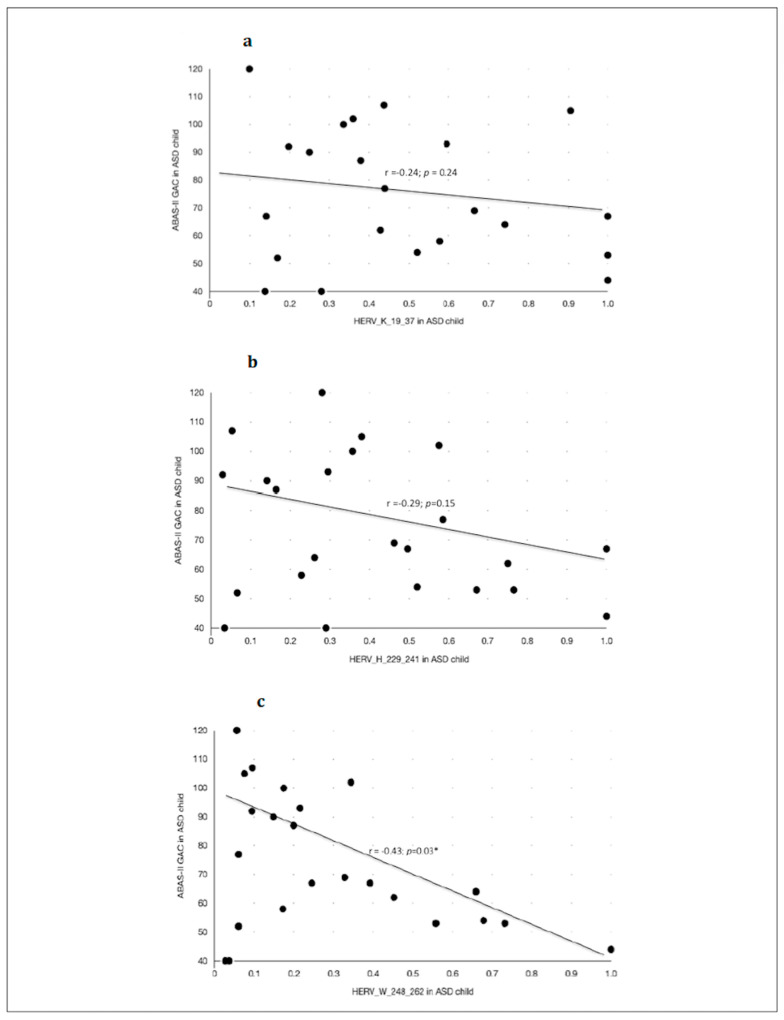
Caption title. Scatter plots of the Pearson’s correlation test between the ABAS-II questionnaire—GAC (Global Adaptative Composite)—Score and the anti-HERVs levels in children with ASD. (**a**) Association between anti-HERV-K and the Global Adaptive Composite Score of the ABAS-II questionnaire in children with ASD diagnosis. (**b**) Association between anti-HERV-H and the Global Adaptive Composite Score of the ABAS-II questionnaire in children with ASD diagnosis. (**c**) Association between anti-HERV-W and the Global Adaptive Composite Score of the ABAS-II questionnaire in children with ASD diagnosis. Statistical significant levels fixed at * *p* < 0.05.

**Table 1 biomedicines-10-01365-t001:** Inclusion and exclusion criteria of the study cohort.

Inclusion Criteria		
ASD Cases	Control Children	Mothers
ASD diagnosis according to DSM-5 Any language and cognitive level Age between 2.5 and 18 years Living, healthy and consanguineous mother	Neurotypical subjects, absence of neuropsychiatric disorders Matched for sex and age with ASD children Living, healthy and consanguineous mother	Neurotypical subjects, absence of neuropsychiatric disorders; Matched for sex and age with the ASD and controls mothers
Exclusion Criteria		
ASD and Control Children		Mothers
Age < 2.5 or >18 years Significant peri- and postnatal complications Malformations or syndromes Significant EEG alteration (except photosensitivity) Significant MRI alterations (e.g., gliosis, hypomyelination); common variants are included Chronic autoimmune and immune-mediated diseases Endocrine diseases (including thyroidopathy, diabetes) Congenital deafness Ongoing infections		Drug, tobacco (>10 cigarettes/day), alcohol use, other teratogens (e.g., radiations) during gestation Gestational diseases (diabetes, infections) chronic autoimmune and immune-mediated diseases (e.g., CD, T1D, SLE, RA, MS), particularly if under specific treatment) Endocrine diseases (including thyroidopathy, diabetes) Ongoing infections

Abbreviations: EEG, electroencephalography; MRI, magnetic resonance imaging; CD, coeliac diseases; T1D, type 1 diabetes; SLE, systemic lupus erythematous; RA, rheumatoid arthritis; MS, multiple sclerosis.

**Table 2 biomedicines-10-01365-t002:** Position and amino acid sequence of the epitopes utilised in the indirect ELISA assay.

	Epitope Position	Amino Acid Sequence	UniProtKb Accession Number
HERV-K-env (19–37)	19–37	VWVPGPTDDRCPAKPEEEG	O42043
HERV-W-env (248–262)	248–262	NSQCIRWVTPPTQIV	Q9UQF0
HERV-H-env (229–241)	229–241	LGRHLPCISLHPW	Q9N2J8

**Table 3 biomedicines-10-01365-t003:** Demographic and clinical characteristics of children.

	ASD		HCs		Group Comparisons		
	*n* = 23		*n* = 19				
	*n*	%	*n*	%	*X^2^*	df	*p*
Males	15	65.21	9	47.36	0.72	1.41	0.39
Other Disorders							
Other NDD (SLD, ADHD)	10	43.47	2	10.52	9.24	1.41	0.002 **
Other Psychiatric Disorders (Anxiety, Mood Disorders)	8	34.78	4	21.05	0.40	1.41	0.52
Familiarity (NDDs)							
Other Pathologies	4	17.39	5	26.31	0.34	1.41	0.55
Prematurity	7	30.43	1	5.2	2.79	1.41	0.03
	**ASD**		**HCs**		**ANOVA**		
	**mean**	**SD**	**mean**	**SD**	**F**	**df**	** *p* **
Age (years)	6.5	4.94	6.0	1.41	0.76	1.41	0.38

*X*^2^: Chi-square test with Yates correction. Abbreviations: NDD, neurodevelopmental disabilities; SLD: specific learning disability; ADHD, attention-deficit/hyperactivity disorder; SD, standard deviation. ** *p* < 0.001.

**Table 4 biomedicines-10-01365-t004:** Demographic and clinical characteristics of the mothers.

	ASD_mothers		HCs_mothers		Group Comparisons		
	*n* = 23		*n* = 19				
	*n*	%	*n*	%	*X* ^2^	df	*p*
Previous Abortion	5	21.73	2	10.52	0.30	1.41	0.57
Threatened Miscarriage	8	34.78	2	10.52	0.14	1.41	0.14
Psychiatric Disorders	3	13.04	0	-	-	-	-
	**ASD**		**HCs**		**ANOVA**		
	**mean**	**SD**	**mean**	**SD**	**F**	**df**	** *p* **
Age (years)	43.0	5.75	30.5	0.70	0.83	1.41	0.63

*X*^2^: Chi-square test with Yates correction. Abbreviations: SD, standard deviation.

**Table 5 biomedicines-10-01365-t005:** Clinical standardised characteristics of the two children group (ASD and controls).

	ASD		HCs		ANOVA		
	Mean	SD	Mean	SD	F	df	*p*
Age (years)	6.5	4.94	6.0	1.41	0.76	1.41	0.38
Adaptive Abilities							
ABASII_GAC	73.73	47.37	90.63	8.48	4.72	1.34	0.03 *
ABASII_DAC	80.39	46.66	97.62	14.84	5.98	1.34	0.02 *
ABASII_DAS	77.86	30.40	95.00	4.94	6.03	1.34	0.01 *
ABASII_DAP	70.30	43.84	90.63	9.89	6.10	1.34	0.01 *
Social Communication Skills							
CGI-S	3.00	1.11	1.42	0.69	55.15	1.41	0.00001 ***
C-GAS	62.0	9.19	89.0	6.43	122.78	1.41	0.00001 ***

Abbreviations: SD, standard deviation. * *p* < 0.05; *** *p* < 0.0001.

**Table 6 biomedicines-10-01365-t006:** Linear regression between all the ages and HERV levels.

Age of All Participants (Children and Mothers)	β [95% CI]	df	*p*
HERV-H-env (229–241)	0.068 [−0.001, 0.38]	1.82	0.41
HERV-W-env (248–262)	1.341 [−0.001, 0.29]	1.82	0.25
HERV-K-env (19–37)	1.004 [−0.001, 0.43]	1.82	0.31

Abbreviations: CI, confidence interval.

**Table 7 biomedicines-10-01365-t007:** ANOVA group comparisons of anti-HERV-env levels between mothers and children of both groups.

	ASD		HCs		ANOVA		
	*n* = 23		*n* =19				
	Mean	SD	Mean	SD	F	df	*p*
child HERV-H-env (229–241)	0.040	0.01	0.44	0.05	0.33	1.41	0.56
mother HERV-H-env (229–241)	0.037	0.35	0.27	0.04	0.08	1.41	0.77
child HERV-W-env (248–262)	0.060	0.04	0.58	0.58	0.02	1.41	0.87
mother HERV-W-env (248–262)	0.320	0.34	0.07	0.02	0.97	1.41	0.32
child HERV-K-env (19–37)	0.036	0.11	0.57	0.33	0.08	1.41	0.76
mother HERV-K-env (19–37)	0.490	0.37	0.31	0.30	0.22	1.41	0.63

Antibody levels were evaluated by indirect ELISA, and the optical density (OD) values were used to determine mean and standard deviation (SD).

**Table 8 biomedicines-10-01365-t008:** Correlation analysis between anti-HERV-env antibody levels and adaptive abilities and social competences measured with ABAS-II (GAC index) in children with ASD and neurotypical HCs.

	ABAS-II GAC			
	ASD		HCs	
	*n* = 23		*n* = 19	
	r	*p*	r	*p*
HERV-H-env (229–241)	−0.29	0.15	0.09	0.670
HERV-W-env (248–262)	−0.43	0.03 *	0.07	0.760
HERV-K-env (19–37)	−0.24	0.24	0.19	0.42

* *p* < 0.05.

## Data Availability

Data supporting reported results can be obtained by asking the corresponding author.

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
