# Peer review of "Antihuman Endogenous Retrovirus Immune Response and Adaptive Dysfunction in Autism"

_biomedicines, 2022, doi:10.3390/biomedicines10061365_

Round 1

Reviewer 1 Report

Carta et al. present analysis of immune responses against human endogenous retroviruses (HERVs) in a cohort of healthy and autistic (ASD) people. Antibodies against envelope proteins of three HERV subgroups (K, H, W) are measured. No significant difference was detected, but there is significant inverse correlation inside ASD cases, between HERV-W antibody levels and adaptive and social competences as measured by dedicated test. That is a relatively minor result, but the authors discuss it in proper way and provide hypotheses about ERV effects on autommunity and neural development.

Comments:

  1. methods, line 136 ... peptides used to generate antibodies are described without sequence details. What specific e.g.  HERV-K sequence was used - specific provirus, consensus sequence?
  2. tables should be better formatted, the current form is way too large
  3. I would just suggest discussing HERVs in the context of autoimmunity as very special antigens. On one hand they are part of "self" proteome, on the other hand they can be handled as "nonself" if not presented in thymic selection.
  4. 4. Line 285-6. Sentence "In typical..." is not clear.

Author Response

REPLY TO REVIEWER 1

Major comments:

  1. "Methods, line 136 ... peptides used to generate antibodies are described without sequence details. What specific e.g. HERV-K sequence was used – specific provirus, consensus sequence?"

Response: The epitopes used in these works derived from the surface domain of the envelope protein of HERV-K, HERV-H and HERV-W. A table containing the information related to each peptide has been added to the manuscript as well (Line 144). The table below shows the aminoacidic sequences of each peptide and the UniProtKb accession number from which the peptide has been obtained.

Epitope position

Aminoacidic sequence

UniProtKb accession number

HERV-K-env (19–37)

19-37

VWVPGPTDDRCPAKPEEEG

O42043

HERV-W-env (248–262)

248-262

NSQCIRWVTPPTQIV

Q9UQF0

HERV-H-env (229-241)

229-241

LGRHLPCISLHPW

Q9N2J8

  1. "Tables should be better formatted, the current form is way too large".

Response: We agree with the reviewer, and we thank you for the suggestion. We changed the previous table formats.

  1. "I would just suggest discussing HERVs in the context of autoimmunity as very special antigens. On one hand they are part of "self" proteome, on the other hand they can be handled as "nonself" if not presented in thymic selection".

Response: In accordance with your suggestion, we have added the following sentences (Line 290-5): It is noteworthy to underline the deceitful nature of HERV antigens (Ags). Theoretically, the host’s immune system should recognize HERVs Ags as “self-Ag” since they are part of the human genome and thus, immune-tolerated. Notwithstanding, HERVs Ags are able to trigger both innate and adaptative responses, probably because of their similarity with exogenous viral proteins.

  1. "Line 285-6. Sentence "In typical..." is not clear".

Response: We agree with the lack of clarity in the Line 285-6 and therefore we have decided to remove it by increasing its clarity in answer number 3.

Reviewer 2 Report

The role of HERV in the autism spectrum disorder is even less clear. The possibility of diagnosing the severity of the disease is very relevant. However, the authors also state that their selection and screening criteria for the patient group showed very little difference compared to the control group. Since it was found that anti-HERV-W antibody levels could be associated with a stress response that could even act as a biomarker, the inclusion of a high stress group in the study could have further confirmed this finding.

Author Response

REPLY TO REVIEWER 2 (in yellow field in the manuscript)

"The role of HERV in the autism spectrum disorder is even less clear. The possibility of diagnosing the severity of the disease is very relevant. However, the authors also state that their selection and screening criteria for the patient group showed very little difference compared to the control group. Since it was found that anti-HERV-W antibody levels could be associated with a stress response that could even act as a biomarker, the inclusion of a high stress group in the study could have further confirmed this finding"

Response: This is pertinent observation and we thank the reviewer for it. We acknowledged the relevance of the suggestion in the Conclusion section: "Our outcomes encourage a replication study to co-evaluate the HERVs gene expression and, since anti-HERV-W antibody levels could be associated with a stress response, with the inclusion of a high stress, non ASD control group, through a follow-up study design"